# Real World Use of Tixagevimab/Cilgavimab Pre-Exposure Prophylaxis of COVID-19 in Immunocompromised Individuals: Data from the OCTOPUS Study

**DOI:** 10.3390/vaccines12070784

**Published:** 2024-07-17

**Authors:** Alessandra Vergori, Giulia Matusali, Eleonora Cimini, Licia Bordi, Paola Borrelli, Simone Lanini, Roberta Palazzi, Jessica Paulicelli, Davide Mariotti, Valentina Mazzotta, Stefania Notari, Rita Casetti, Massimo Francalancia, Silvia Rosati, Alessandra D’Abramo, Cosmina Mija, Paola Mencarini, Eugenia Milozzi, Emanuela Caraffa, Simona Sica, Elisabetta Metafuni, Federica Sorà, Angela Rago, Agostina Siniscalchi, Elisabetta Abruzzese, Mariagrazia Garzia, Giovanni Luzi, Roberta Battistini, Luca Prosperini, Antonella Cingolani, Enrico Girardi, Fabrizio Maggi, Andrea Antinori

**Affiliations:** 1Viral Immunodeficiencies Unit, National Institute for Infectious Diseases “L. Spallanzani”, Istituto di Ricovero e Cura a Carattere Scientifico (IRCCS), 00149 Rome, Italy; simone.lanini@uniud.it (S.L.); jessica.paulicelli@inmi.it (J.P.); valentina.mazzotta@inmi.it (V.M.); andrea.antinori@inmi.it (A.A.); 2Laboratory of Virology, National Institute for Infectious Diseases “L. Spallanzani”, Istituto di Ricovero e Cura a Carattere Scientifico (IRCCS), 00149 Rome, Italy; giulia.matusali@inmi.it (G.M.); licia.bordi@inmi.it (L.B.); davide.mariotti@inmi.it (D.M.); massimo.francalancia@inmi.it (M.F.); cosmina.mija@inmi.it (C.M.); fabrizio.maggi@inmi.it (F.M.); 3Laboratory of Cellular Immunology and Pharmacology, National Institute for Infectious Diseases “L. Spallanzani”, Istituto di Ricovero e Cura a Carattere Scientifico (IRCCS), 00149 Rome, Italy; eleonora.cimini@inmi.it (E.C.); stefania.notari@inmi.it (S.N.); rita.casetti@inmi.it (R.C.); 4Laboratory of Biostatistics, Department of Medical, Oral and Biotechnological Sciences, University “G. D’Annunzio” of Chieti-Pescara, 66100 Chieti, Italy; paola.borrelli@unich.it; 5Clinica Malattie Infettive, Department of Medicine, University of Udine, 33100 Udine, Italy; 6Accettazione e Teleconsulto Rete Regionale Malattie Infettive, National Institute for Infectious Diseases “L. Spallanzani”, Istituto di Ricovero e Cura a Carattere Scientifico (IRCCS), 00149 Rome, Italy; roberta.palazzi@inmi.it; 7Emerging Infectious Diseases Unit, National Institute for Infectious Diseases “L.Spallanzani”, Istituto di Ricovero e Cura a Carattere Scientifico (IRCCS), 00149 Rome, Italy; silvia.rosati@inmi.it (S.R.); alessandra.dabramo@inmi.it (A.D.); 8Respiratory Infectious Diseases Unit, National Institute for Infectious Diseases “L. Spallanzani”, Istituto di Ricovero e Cura a Carattere Scientifico (IRCCS), 00149 Rome, Italy; paola.mencarini@inmi.it; 9Epatology Unit, National Institute for Infectious Diseases “L. Spallanzani”, Istituto di Ricovero e Cura a Carattere Scientifico (IRCCS), 00149 Rome, Italy; eugenia.milozzi@inmi.it; 10Severe and Immune-Depression Associated Infectious Diseases Unit, National Institute for Infectious Diseases “Lazzaro Spallanzani”, Istituto di Ricovero e Cura a Carattere Scientifico (IRCCS), 00149 Rome, Italy; emanuela.caraffa@inmi.it; 11Imaging Diagnostics Departmenti, Radioterapia Oncologica e Ematologia, Fondazione Policlinico Universitario Agostino Gemelli, IRCCS, 00184 Rome, Italy; simona.sica@policlinicogemelli.it (S.S.); elisabetta.metafuni@policlinicogemelli.it (E.M.); federica.sora@policlinicogemelli.it (F.S.); 12UOSD Ematologia ASL Roma 1, San Filippo Neri Hospital, 00135 Rome, Italy; angela.rago@aslroma1.it (A.R.); agostina.siniscalchi@aslroma1.it (A.S.); 13UOC Ematologia, Sant’Eugenio Hospital, 00144 Rome, Italy; elisabetta.abruzzese@uniroma2.it; 14UOC Ematologia e Trapianto di Cellule Staminali, San Camillo-Forlanini Hospital, 00152 Rome, Italy; mgarzia@sancamilloforlanini.rm.it (M.G.); gluzi@sancamilloforlanini.rm.it (G.L.); rbattistini@sancamilloforlanini.rm.it (R.B.); 15UOC Neurologia e Neurofisiopatologia, San Camillo-Forlanini Hospital, 00152 Rome, Italy; luca.prosperini@gmail.com; 16Infectious Diseases Institute, Policlinico A. Gemelli, 00136 Rome, Italy; antonella.cingolani@policlinicogemelli.it; 17Scientific Direction, National Institute for Infectious Diseases “L. Spallanzani”, IRCCS, 00184 Rome, Italy; enrico.girardi@inmi.it; 18Clinical Department and Research Direction, National Institute for Infectious Diseases “L. Spallanzani”, IRCCS, 00184 Rome, Italy

**Keywords:** passive pre-prophylaxis, SARS coronavirus, cell mediated immunity, humoral immunity

## Abstract

**Objective**. We aimed to report the real-world use and outcomes over time in immunocompromised individuals receiving tixagevimab/cilgavimab (T/C) pre-exposure prophylaxis (PrEP). **Methods**. This observational study included participants who received T/C PrEP, categorized into three groups: (i) No COVID-19 (NoC), i.e., participants who never had COVID-19; (ii) Hybrids (H), i.e., participants who had COVID-19 before PrEP; and (iii) Break-through Infections (BTIs), i.e., participants who had COVID-19 after PrEP. The study measured several immune markers at the administration of T/C (T0) at 3 (T1), 6 (T2), and 9 (T3) months afterward. These markers included: anti-receptor-binding domain (RBD) IgG antibodies; BA.5-neutralizing antibodies (nAbs); mucosal IgG; and T cell immunity. The incidence rate ratios for BTIs were analyzed using a Poisson regression model. **Results**. A total of 231 participants with a median age of 63 years (IQR 54.0–73.0). were included. Among these, 84% had hematological diseases and received a median of three vaccine doses. N = 72 participants belonged to the NoC group, N = 103 to the H group, and n = 56 to the BTI group (24%), with most BTIs being mild/moderate. The incidence rate (IR) of BTIs was 4.2 per 100 patient-months (95% CI 3.2–5.4), with no associated risk factors identified. There was a significant increase in anti-RBD IgG levels 3 months after the T/C administration in all groups, followed by a decline at 6 months, whereas at the same time points, geometric mean titers (GMTs) of anti-BA.5 nAbs were low for all groups and were around or below the detection threshold. No significant changes were observed in IFN-γ levels. The mucosal immune response was observed only 3 months after the PrEP administration. **Conclusion**. We provided a real-world experience model on the clinical efficacy of T/C PrEP in preventing severe COVID-19 during the Omicron wave through a comprehensive virological and immunological study. While waiting for the arrival of new monoclonal antibodies that can effectively neutralize the most recent variants, T/C PrEP remains the only viable strategy in the available armamentarium today to prevent COVID-19 complications in an extremely fragile population with suboptimal immune responses to COVID-19 vaccines.

## 1. Introduction

Immunocompromised individuals may experience reduced vaccine immune response with an impaired seroconversion and effectiveness [1,2,3], and are at higher risk for severe coronavirus disease 2019 (COVID-19) and death, especially those with hematological malignancies and organ transplant recipients [4,5,6,7,8].

To address the need to protect them from breakthrough infections (BTIs) and possibly long-lasting severe acute respiratory syndrome coronavirus 2 (SARS-CoV-2) infections, in December 2021, the combination of tixagevimab/cilgavimab (Evusheld^TM^, AstraZeneca, T/C) received emergency use authorization (EUA) from the United States Food and Drug Administration (FDA) as pre-exposure prophylaxis (PrEP). T/C PrEP was approved for individuals aged 12 years or older (weighing at least 40 kg) who were either are moderate to severely immunocompromised or had an inadequate response to SARS-CoV-2 vaccination [9]; subsequently, it was approved in Europe and in March 2022 in Italy at a dosage of 150/150 mg given intramuscularly (IM) [9,10,11,12]. It became a mainstay for protecting immunocompromised patients, despite evidence regarding virological partial efficacy against newer Omicron sublineages [13,14,15,16,17].

The current recommended and revised dosing for PrEP by the United States FDA is 300 mg/300 mg of T/C to ensure the neutralization of a combination of monoclonal antibodies (mAbs) against the COVID-19 Omicron sublineages. It is administered as separate sequential intramuscular injections [9], but this dosage revision has not been implemented by Europe or Italy.

This combination of monoclonal antibodies was found to be fully effective against Omicron variant B.1.1.529 with Evusheld’s inhibitory concentrations 50 (IC50s) (a measure of neutralizing potency) of 171 ng/mL and 277 ng/mL in two confirmatory tests. This is within the range of neutralizing titers found in someone who has previously been infected with COVID-19. Evusheld’s IC50s for the original strain of SARS-CoV-2, previously referred to as the Wuhan strain, were approximately 1.3 ng/mL and 1.5 ng/mL, respectively [18].

To date, there have been several studies on the efficacy of this mAbs combination, showing reduced rates of risk of hospitalization from 53% to 100%, an 87–100% reduction in the risk of ICU admission, and a 64–100% reduction in deaths in immunocompromised individuals [19,20,21,22,23], including those with hematological malignancies [24,25,26,27] and solid organ recipients (SOT) [28], in the Omicron era.

Interestingly, a recent review and meta-analysis of 30 studies and 27,932 participants showed that T/C use as PrEP was associated with lower COVID-19-related hospitalization, ICU admission, and mortality rates; a lower proportion of patients who needed oxygen therapy; a lower RT-PCR SARS-CoV-2 positivity rate; and a lower proportion of patients who had severe COVID-19 compared to those not receiving treatment or alternative treatment for the prevention of COVID-19 [29].

Real-world effectiveness (RWE) studies have reported variable percentages of COVID-19 infection after the administration of T/C PrEP, ranging from 4% of BTIs in predominance of the Delta, BA.1, and BA.2 Omicron sublineages [21,25] to 11–46% in predominance of the BA.5 sublineage [20,29,30].

This evidence shows that the use of T/C in PrEP is substantially dependent on the current epidemiological context and circulating variants, as the circulation of newer variants leads to conflicting in vitro results with reduced neutralizing activity, a useful tool to assess whether a given monoclonal antibody will be effective for a variant.

Concerning this, a few in vitro studies have demonstrated the neutralizing activity of T/C when used as PrEP against the most recent variants of concern (VOCs) or interest (VOIs), and the results are controversial. Some claim that T/C has and maintains neutralizing activity against Delta, BA.2, and BA.5 even at 6 months after PrEP administration [13]; others that neutralization against BA.5 poses similar challenges to BA.1 in light of their similar immune escape profiles [31]. This evidence and the preponderant circulation of new Omicron sublineages prompted the FDA to discontinue the use of T/C for pre-exposure prophylaxis treatment, withdrawing the emergency use authorization [32].

However, the European Medicines Agency (EMA) did not take the same action. The use of T/C was paused for treatment, but not for pre-exposure prophylaxis.

Herein, we report the clinical efficacy, antibody persistence, neutralizing activity, and mucosal and specific T immunity over time in severely immunocompromised patients undergoing PrEP with T/C.

## 2. Materials and Methods

On 1 March 2022, the National Institute for Infectious Diseases Lazzaro Spallanzani in Rome, Italy, started an outpatient service for COVID-19 PrEP according to the recommendations of the Ministry of Health and Italian National Regulatory Agency, using a combination of tixagevimab/cilgavimab (T/C) (Evusheld^TM^, AstraZeneca) at a dosage of 150/150 mg intramuscularly (IM) in immunocompromised individuals unable to mount any humoral response to SARS-CoV-2 infection or vaccine at least 4 months afterward, who are at higher risk of severe COVID-19 [11,12]. The OCTOPUS study is an observational study on the outcomes of PrEP with T/C. According to the protocol, the demographic, epidemiologic, clinical, and laboratory characteristics of the participants were collected. The main study outcomes were the persistence over time of the main immune markers (anti-receptor-binding domain/spike IgG, anti-RBD/S IgG) of T/C. The study evaluated the clinical outcomes of prophylaxis with T/C in terms of the incidence of SARS-CoV-2 infections, including severe/critical manifestations and deaths within 12 months following PrEP administration, and the incidence rate of serum anti-Nucleocapside (anti-N) IgG conversion in consistently asymptomatic subjects, as well as its safety. Only in a subgroup of participants was the objective to assess the persistence over time of the neutralizing activity to circulating VOCs or VOIs. This was intended to determine the presence and persistence of T/C concentrations in anatomical compartments other than the blood, such as upper airway mucosa (saliva), and to assess the specific anti-SARS-CoV-2 cellular T response following T/C administration in the study population. By protocol, following written informed consent, blood samples were collected for all participants enrolled at the time of administration (baseline, T0), 3-, 6-, 9- and 12 months after administration of T/C (T1, T2, T3, T4). Saliva samples were collected for all participants after 3-, 6- and 9-months following T/C administration (Appendix A). The Scientific Committee of the Italian Drug Agency and the Ethical Committee of the Lazzaro Spallanzani Institute, as the national review board for the COVID-19 pandemic in Italy, approved the study (approval number: 80/2022).

In this analysis, we present results on anti-RBD IgG persistence, neutralizing activity, and cell-mediated and mucosal immunity at 3, 6, and, when available, 9 months of follow-up. The study population consisted of those participants who received T/C for PrEP at a dosage of 150/150 mg IM between March 2022 and March 2023, stratified according to immunization status: those who were previously vaccinated and never had COVID-19 (No COVID-19; NoC); those who were previously vaccinated and had COVID-19 before PrEP (Hybrids; H); and those who were previously vaccinated and had COVID-19 after PrEP (breakthrough infections; BTIs).

### 2.1. Laboratory Procedures

SARS-CoV-2-specific anti-RBD IgG and anti-N IgG were measured using two commercial chemiluminescence microparticle antibody assays (CMIA; ARCHITECT SARS-CoV-2 IgG; and ARCHITECT SARS-CoV-2 IgG II Quantitative, Abbott Laboratories, Wiesbaden, Germany, respectively) performed on ARCHITECT^®^ i2000sr (Abbott Diagnostics, Chicago, IL, USA) and used according to the manufacturers’ instructions. Index > 1.4 and binding antibody units (BAU)/mL ≥ 7.1 were considered positive, respectively. At each time point, SARS-CoV-2 Omicron BA.5-neutralizing antibodies (nAbs) were measured using a live-virus microneutralization assay (MNA90), as previously described [33].

Saliva samples were self-collected at each time point under medical personal supervision by passive drooling. They were spontaneously produced without external stimuli, placed into sterile containers without any buffer added, centrifuged at 12,000 rpm for 5 min, aliquoted, and stored at −20 °C. Indirect immunofluorescence assay (IFA) was performed to evaluate the mucosal anti-SARS-CoV-2-specific IgG and IgA titers using saliva samples serially diluted from 1:2 down to 1:128 incubated with anti-human IgA and IgG rabbit antibodies, then conjugated with FITC and counterstained with Evans Blue (Euroimmun, Lubecca, Germany). Preparation of in-house slides was performed in a BSL-3 laboratory using Vero E6 cells infected with SARS-CoV-2 isolate (SARS-CoV-2/Human/ITA/PAVIA10734/2020) and fixed with cold acetone. The T cell-specific response to SARS-CoV-2 spike protein (peptides 1 ug/mL; Miltenyi) was assessed according to IFN-γ production by an ELISA assay (Biotechne).

### 2.2. Statistical Analysis

Descriptive analysis was carried out using the mean and standard deviation (SD) or median and interquartile range (IQR) for the quantitative variables and percentage values for the qualitative ones. Normality distribution for quantitative variables was assessed using the Shapiro–Wilk Test. Pearson’s chi-square test or Fisher’s exact test were used to evaluate the association between categorical variables, while the non-parametric Kruskal–Wallis test was used between continuous variables and the considered groups. After the Kruskal–Wallis test, for the statistically significant results, the Dunn test was calculated compare between the pairs of medians for the identification of significant differences. Also, the Friedman test was used to determine the differences between the medians of time points for laboratory variables. For significant trends, this analysis was followed by the Sign test. The Bonferroni’s correction for multiple comparison tests was applied.

In addition, the incidence rate per 100 patient-months and 95% confidence intervals of BTIs were estimated, and a Poisson regression model was utilized to calculate specific BTI incidence rate ratios (IRRs) and 95% confidence intervals, considering age, gender, number of vaccine doses received, anti RBD IgG, BA.5 nAbs, and IFN-γ at T0.

Statistical significance was set at the level of ≤0.05 unless adjustment for multiple comparisons was needed. All analyses were performed using Stata software v17.1 (StataCorp, College Station, TX, USA).

## 3. Results

### 3.1. Study Population

We enrolled 231 patients with a median age of 63.0 years (interquartile range (IQR) 54.0–73.0), 46% of which were female, 84% of which had a hematological disease (44%: non-Hodgkin lymphoma, 25%: multiple myeloma, 12%: chronic lymphocytic leukemia), 33% of which had received anti-CD20 treatment, and 3% of which had underwent CAR-T. Of those who had non-hematological diseases (16%, 37/231), 17 were represented by multiple sclerosis, 6 by solid organ transplant, and 1 by advanced HIV, while 13 had been treated with immune-suppressive therapy for autoimmune disorders. In addition, 57% had at least one comorbidity (details in Table 1). The median vaccine doses received was three. The general characteristics of the study population and their immunization statuses are reported in Table 1. As for the study groups, N = 72/231 had no COVID-19 during the observational follow-up period, 103/231 had had COVID-19 before PrEP, and 56/231 (24%) were diagnosed with BTIs. The three groups did not differ in their general characteristics, except for the presence of hematological diseases, which was more frequently found in the NoC and BTIs groups (90% and 87%, respectively, versus 78% in H; *p* = 0.058) and for COPD, which was mostly found in the NoC group (7% vs. 0% in BTIs and 1% in H; *p* = 0.028). No adverse events due to T/C were observed.

### 3.2. Anti-RBD IgG and BA.5-Neutralizing Antibodies Persistence

At the time of administration (T0) and in 219 out of 231 available samples, the median titer of anti-RBD IgG was 212.7 BAU/mL (5.9–979.0). According to the study groups, titers were significantly lower in the BTIs group compared to the H group (26.1 (1.4–360.5) vs. 474.1 (105.1–1607.5); *p* < 0.001) (Table 2). No differences were observed among the three groups at any of the time points (Table 2 and Figure 1A). Overall, the anti-RBD IgG levels significantly increased from T0 to T1 (*p* < 0.001), with an overall median value of 1589.1 BAU/mL (1147.2–2485.3) (Table 2). However, a significant decrease was observed at each subsequent time point with overall medians of 666.0 (478.6–1166.7) at T2 and 277.8 (182.2–459.1) at T3 (*p* < 0.001). Antibody kinetics were similar across the three groups, and after administration, anti-RBD IgG levels remained above the assay positivity threshold until T3 (Figure 1A).

Regarding nAbs against BA.5, in 142 out of 231 available samples at the time of PrEP administration, mean titers below the cut-off were observed, with a statistically significant difference due to a wider IQR in the H group compared to the NoC and BTIs groups (*p* < 0.001) (Table 2). These titers significantly increased from T0 to T1 (*p* < 0.001) to a geometric mean titer (GMT) of 19.6 (15.9–23.1), with no differences among the groups. There was a significant decrease at T2 from T1, returning to the cut-off limits. At T2, although the values were near the assay cut-off, higher titers of BA.5 nAbs were found in the H group compared to the NoC group (11.7 (8.2–16.7) vs. 5.6 (5.1–6.2); *p* = 0.002), as well as in BTIs compared to the NoC group (13.4 (7.7–23.3) vs. 5.6 (5.1–6.2); *p* = 0.006). At T3, higher titers in BTIs [17.4 (8.9–33.7)] were found compared to both the NoC group (5.2 (4.8–5.6); *p* < 0.001) and the H group (5.0 (5.0–5.0); *p* = 0.002). Despite these statistical significances, the mean titers of nABs remained below or near the threshold limit throughout all the examined time points (Table 2 and Figure 1B).

### 3.3. Mucosal Anti- SARS-CoV-2 IgG Persistence

Concerning the kinetics of IgG at the mucosal level, a similar trend was observed in all three studied groups, with the mucosal antibody levels remaining mostly constant between T1 and T2 and declining thereafter at T3, but not in a significant manner. Noteworthily, a significant difference was observed between T1 and T3 in both the NoC (geometric mean: 2.46 (1.74–3.48) vs. 1.13 (0.98–1.32); *p* = 0.032) and the H groups (geometric mean: 2.76 (2.11–3.57) vs. 1.20 (0.79–1.84); *p* = 0.025), while in BTI, the difference was not significant (geometric mean: 2.77 (1.89–4.06) vs. 2.26 (1.13–4.51); *p* = 0.830), with the antibody levels remaining above the positivity threshold (Figure 2). Unfortunately, regarding the saliva samples, we have no data concerning T0, since the collection of this biological fluid started from T1. Moreover, the saliva samples which tested positive for IgG in BTIs were also analyzed for the presence of IgA; the results highlight that 50% of IgG-positive samples also showed the simultaneous presence of mucosal IgA (51.8% at T1, 40% at T2, and 55.5% at T3) (Appendix A). The IgA mucosal antibody levels remained mostly constant and slightly above the positivity threshold at both T1 (geometric mean: 2.65 (1.64–4.27)) and T2 (geometric mean: 2.29 (1.17–4.48)); higher mean levels were observed at T3 (geometric mean: 4 (1.25–12.78)).

### 3.4. T-Specific Cell Response Assessed by IFN-γ Release

Overall, in 149/231 available samples, a median of 38.2 pg/mL (7.0–204.4) before PrEP was found, which is above the cut-off, with significantly higher levels in H than NoC (92.0 (16.1–343.4) vs. 27.5 (4.0–140.0) in NoC; *p* = 0.014). After the administration of PrEP, no significant changes in median levels were found, with no differences among the groups (Table 2 and Figure 3).

### 3.5. Breakthrough Infections (BTIs)

A total of 56 out of 231 (24%) participants experienced a BTI after PrEP. They had a median age of 63 years (SD + 11.5), 52% were males, 87% had a hematological disease, 68% had at least one comorbidity, and 82% had received three vaccine doses before receiving PrEP. Of these BTIs, 9% were asymptomatic, 55.3% were mild, 10.7% were moderate, and only one BTI was reported as severe, resulting in a non-COVID-19-related death. Of the BTIs, 40% were caused by the BA.5 variant, followed by 20% caused by BA.1, 20% by BA.2, and 20% by BA.4 (based on available sequencing). The laboratory parameters, similarly to the overall population and to H and NoC, showed significant increases in anti-RBD IgG and BA.5 nAbs titers from T0 to T1 (Table 2). On the contrary, the subsequent decreases in the same parameters at T2 did not reach statistical significance for either anti-RBD IgG or BA.5 nAbs titers (Table 2). At T3, the BTIs population was the only one showing mean BA.5 nAbs titers above the limit of the assay detection 17.4 (8.9–33.7). No changes in IFN-γ production were observed. The IR was 4.2 (95%CI 3.2–5.4) BTIs/100 patient-months, and no factors associated with an increased or reduced IR were found among those analyzed (age, gender, number of vaccines doses received, anti-RBD IgG, BA.5 nAbs, or IFN-γ at T0). Moreover, 46% occurred within T1 (within 3 months after PrEP), 37% within T2 (within 6 months), and 17% within T3 (within 9 months).

## 4. Discussion

This study shows 24% of BTIs in immunocompromised subjects under T/C PrEP with an IR of 4.2 BTIs/100 patient-months, mainly mild/moderate and with no related deaths. The proportion of BTIs found was relatively high compared to other previously reported results in the Omicron era. Rates of BTIs of around or less than 10% have been reported in several studies after times of follow-up of fewer than or equal to three months. Indeed, a large study of immunocompromised patients with a median follow-up of 63 days found a proportion of BTIs of 4.4% [21], and 6% was found in a mixed cohort in which most participants had hematologic diseases after a median follow-up time of 48 days [23]. A similar finding was reported by another study where only 3.5% of participants were infected with SARS-CoV-2, with no deaths attributed to COVID-19 [22]. Our percentage also seems to be a higher rate than that of BTIs found in patients with B cell malignancies, in which, after a median time of 91 days from T/C administration, 11% experienced BTIs [27]. It is also higher than that observed in multiple myeloma-affected patients during the follow-up period of a median of 31 days (8.1%), with no SARS-CoV-2-infection-related hospitalizations or deaths [34], and than 9.3% of BTIs after an average time of follow-up of 151 days in a large cohort of immunocompromised subjects [24].

Referring to BTI in terms of IR, our result appears to be close to that found in a propensity scores-matched analysis conducted in Israel on a large cohort of immunocompromised patients, in which 10% of BTIs were found, reflecting an incidence rate of 4.18 per 100 person-months [35]. Moreover, this estimate seems to be consistent, since the Poisson model was adjusted for number of vaccine doses received and anti-RBD-IgG; thus, interference of these adjustment variables is unlikely.

In light of all this evidence, we must point out that the previously reported BTI rates are heterogeneous, making the results hardly comparable; some studies have shown beneficial effects of T/C PrEP use, while others have reported low neutralizing capabilities and non-significant results against the most recent Omicron subvariants. Others stated that it depends on T/C dosage, although a statistically significant difference between the incidence of COVID-19 infections and patients receiving a single 150/150 mg dose compared with those receiving a cumulative 300/300 mg first dose has not been found [26].

Moreover, most of these previously published works were conducted in epidemiological eras in which early Omicron subvariants, namely, BA.1 and BA.2, prevailed, and lower rates of BTIs and severe COVID-19 during that Omicron wave have been found [18]. Finally, some studies relied on higher T/C PrEP dosages, making the results even less comparable.

The measured anti-RBD IgG titers tended to decrease significantly at 6 months, confirming what has already been demonstrated regarding the pharmacokinetic characteristics of the T/C combination: their mean half-life is approximately 90 days for both mAbs, with a reported and ensured neutralizing activity for more than 6 months at a single dosage of 300 mg [34]. However, this is questionable in more recent Omicron eras.

Poisson models did not show any factor associated with an increased or reduced incidence of BTIs, and almost half of the BTIs occurred in the first 3 months after PrEP, suggesting that they are probably dependent on nAbs titers.

Particularly, what emerges are relatively low rates of BTIs in short follow-up times, but considering our follow-up time, which was longer than those of the aforementioned studies, and that PrEP administration mostly occurred in the BA.5 era, a higher rate of BTI can be expected.

Our analysis showed, at 3 months from the administration of T/C, low to undetectable levels of antibodies able to neutralize the BA.5 sublineage, confirming that the current dosage of 150/150 mg might be not sufficient to ensure adequate neutralization of this strain.

The emergence of the Omicron strain and its sub-variants hindered T/C efficacy in PrEP use, as several studies showed decreased potency against them [13,14]. Mixed results were reported regarding monoclonal antibodies’ neutralizing activity against Omicron lineages [15,16]. Bruel et al. demonstrated residual activity of cilgavimab against BA.5 at high dosages [15] and a 10–100-fold reduction in neutralization activity against BA.5 than the neutralizing activity against BA.1 due to T/C, with no activity at all against BQ.1.1 or XBB1.5 [17].

Furthermore, considering that SARS-CoV-2 infection occurs at the upper respiratory tract level and that the first immune response to be induced is a mucosal response, we also evaluated the presence and persistence of IgG at the mucosal level, since a good local antibody response can result in an asymptomatic infection or a disease with mild symptoms [36].

The results obtained from saliva samples showed a similar trend of IgG in all three studied groups, with the mucosal antibody levels remaining mostly constant between T1 and T2 and declining thereafter at T3. Significant differences between T1 and T3 were observed in both the NoC and in H groups, while in BTI, the difference was not significant. Moreover, saliva samples which tested positive for IgG were also analyzed for the presence of IgA; the results obtained for the BTIs group highlighted that, among the 40 samples which were positive for IgG, 50% (20/40) also showed the simultaneous presence of mucosal IgA. Further analyses will be necessary to better clarify this issue, considering that several research groups have demonstrated that natural SARS-CoV-2 infection is efficient in inducing both mucosal IgG and IgA production, with IgA levels often higher than those of IgG [36,37].

Regarding the T cell-specific response, we observed a difference only at baseline between those participants with a hybrid condition (H, previously vaccinated and naturally infected with SARS-CoV-2) and those who did not have COVID-19 (NoC), showing higher levels of IFN-γ in the former condition vs. the latter. This mirrors what is already known about T cell immunity in hybrids, namely, a robust and persistent Omicron-reactive T cell response increases markedly upon booster vaccination, also indicating an additive effect of ancestral-based booster vaccination and Omicron infection on cellular immunity across different immunocompromised states, thus conferring protection against severe COVID-19 [38].

This study has some limitations, including the retrospective and observational design of the study, which can introduce unmeasured and confounding bias. Also, we did not include matched controls, either unexposed or exposed to a higher dosage of T/C, to compare such findings. We were not able to test all the patients for SARS-CoV-2 using real-time reverse transcriptase-polymerase chain reaction (PCR) or by sequencing all samples during the follow-up period, as most participants were diagnosed using rapid antigen tests or PCR performed in out-of-hospital laboratories, which did not provide variant data. Additionally, although none of the participants were in full remission or were off treatment for many months, the exact date of the last immunosuppressive treatment was not available in the database, and this could be a source of residual confounding. Moreover, the test for measuring anti-RBD IgG was unable to distinguish between administered mAbs-, infection-, and vaccine-driven IgG. Nevertheless, the participants in this study had non-responses or weak responses to previous SARS-CoV-2 vaccination or to COVID-19 as an inclusion criterion for PrEP because they were immunocompromised. Finally, saliva samples were collected only starting from three months post-prophylaxis; therefore, data related to T0 were missing for this biological fluid, and it was not possible to compare serological and mucosal IgG at the time of PrEP. The study’s strengths include a large, diverse population of immunocompromised subjects at increased risk of severe COVID-19 and related complications.

## 5. Conclusions

We provided a real-world experience model of the clinical efficacy of T/C PrEP in preventing severe COVID-19 during the Omicron wave, which is reported to have been associated with increased BTIs among vaccinated subjects, jointly with a comprehensive virological and immunological study. In conclusion, while waiting for the arrival of new monoclonal antibodies that can effectively neutralize the most recent variants while also possibly preventing infection, T/C PrEP remains the only viable strategy in the available armamentarium today, preventing COVID-19 complications in an extremely fragile population with suboptimal immune responses to COVID-19 vaccines.

## Figures and Tables

**Figure 1 vaccines-12-00784-f001:**
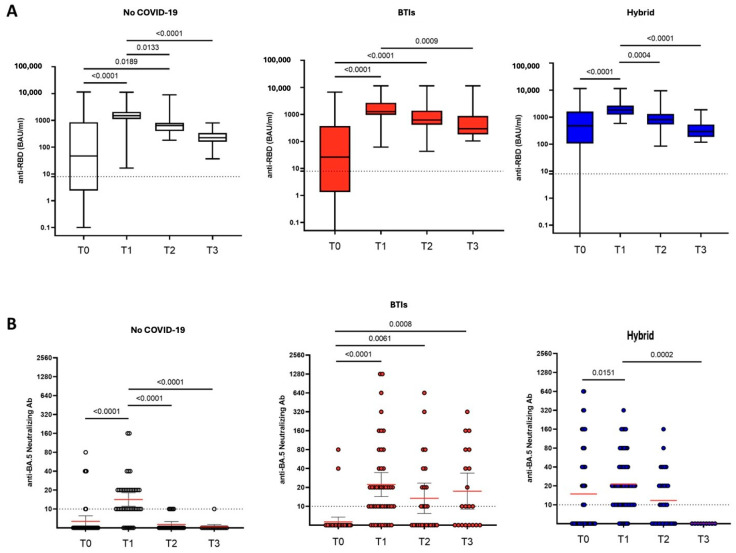
Anti-RBD IgG (**A**) and Omicron BA.5-neutralizing antibody (**B**) levels measured in serum of patients undergoing PrEP. Dot lines represent the cut-offs of the assays. Kruskal–Wallis and Dunn’s tests were performed for statistical comparisons. Red lines in 1B represent geometric mean values.

**Figure 2 vaccines-12-00784-f002:**
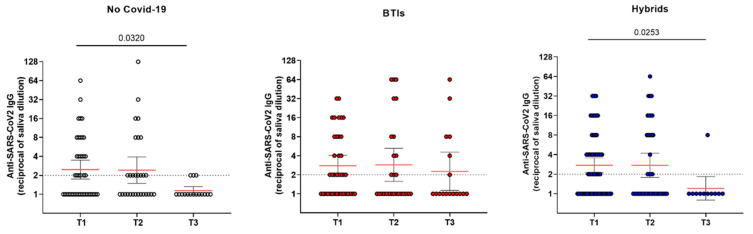
Anti-SARS-CoV-2 IgG levels measured in saliva. Dot lines represent the cut-offs of the assays. Kruskal–Wallis and Dunn’s tests were performed for statistical comparison. Red lines represent geometric mean values.

**Figure 3 vaccines-12-00784-f003:**
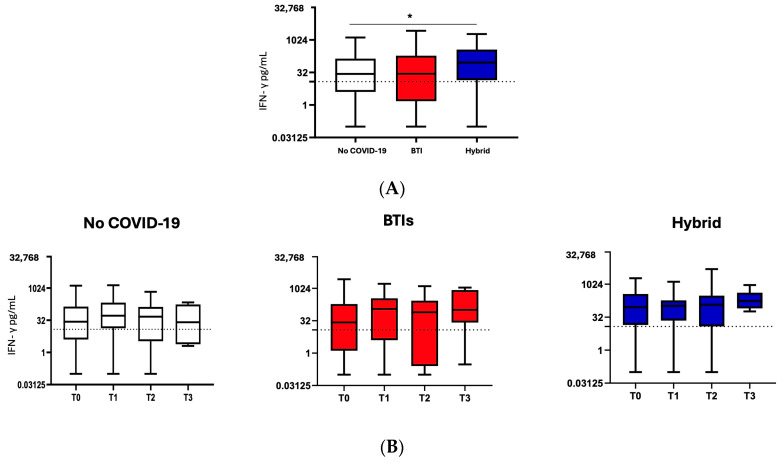
T cell-specific response was assessed by IFN-γ release after SARS-CoV-2 peptide stimulation according to the study groups. Mann–Whitney test was performed for statistical comparison at T0 (**A**) and throughout the study time points (**B**); * *p* = 0.01.

**Table 1 vaccines-12-00784-t001:** General characteristics of study population according to SARS-CoV-2 infection status: No COVID19 (NoC), Breakthrough infections (BTIs), and Hybrids (H).

	Total	No COVID-19 (NoC)	Breakthrough Infections (BTIs)	Hybrid Immunity (H)	
	N = 231	N = 72	N = 56	N = 103	*p*-Value
**Age, median (IQR)**	63.0 (54.0–73.0)	66.0 (54.0–73.0)	65.0 (56.0–71.0)	63.0 (52.0–73.0)	0.630
**Gender, n(%)**					0.931
Male	124 (53.9)	39 (54.2)	29 (51.8)	56 (54.9)	
Female	106 (46.1)	33 (45.8)	27 (48.2)	46 (45.1)	
**Hematological disease, n** **(%)**					0.058
No	37 (16.0)	7 (9.7)	7 (12.5)	23 (22.3)	
Yes	194 (84.0)	65 (90.3)	49 (87.5)	80 (77.7)	
**Types of hematological diseases, n (%)**					0.238
Non-Hodgkin Lymphoma	86 (44.3)	32 (49.2)	24 (49.0)	30 (37.5)	
Multiple Myeloma	49 (25.3)	16 (24.6)	14 (28.6)	19 (23.8)	
Chronic Lymphocytic Leukemia	23 (11.9)	6 (9.2)	2 (4.1)	15 (18.8)	
Others	36 (18.6)	11 (16.9)	9 (18.4)	16 (20.5)	
**Immunosuppressive treatment, n (%)**					0.141
No	16 (7.4)	2 (3.1)	3 (5.9)	11 (11.1)	
Yes	199 (92.6)	63 (96.9)	48 (94.1)	88 (88.9)	
**Anti-CD20 treatment, n (%)**					
No	155 (67.1)	47 (65.3)	39 (69.6)	69 (67.0)	0.872
Yes	76 (32.9)	25 (34.7)	17 (30.4)	34 (33.0)	
**CAR-T, n** **(%)**					0.029
No	223 (97.0)	68 (94.4)	53 (94.6)	103 (100.0)	
Yes	7 (3.0)	4 (5.6)	3 (5.4)	0 (0.0)	
**Comorbidities, n (%)**					0.133
No	100 (43.3)	32 (44.4)	18 (32.1)	50 (48.5)	
Yes	131 (56.7)	40 (55.6)	38 (67.9)	53 (51.5)	
**Hypertension, n (%)**					0.715
No	200 (86.6)	64 (88.9)	47 (83.9)	89 (86.4)	
Yes	31 (13.4)	8 (11.1)	9 (16.1)	14 (13.6)	
**Diabetes, n (%)**					0.526
No	214 (92.6)	67 (93.1)	50 (89.3)	97 (94.2)	
Yes	17 (7.4)	5 (6.9)	6 (10.7)	6 (5.8)	
**Cardiovascular Disease, n (%)**					0.817
No	227 (98.3)	70 (97.2)	55 (98.2)	102 (99.0)	
Yes	4 (1.7)	2 (2.8)	1 (1.8)	1 (1.0)	
**Dyslipidemia, n (%)**					0.320
No	226 (97.8)	69 (95.8)	56 (100.0)	101 (98.1)	
Yes	5 (2.2)	3 (4.2)	0 (0.0)	2 (1.9)	
**Previous cancer, n (%)**					0.358
No	212 (91.8)	66 (91.7)	49 (87.5)	97 (94.2)	
Yes	19 (8.2)	6 (8.3)	7 (12.5)	6 (5.8)	
**COPD, n (%)**					0.028
No	225 (97.4)	67 (93.1)	56 (100.0)	102 (99.0)	
Yes	6 (2.6)	5 (6.9)	0 (0.0)	1 (1.0)	
**Death, n (%)**					0.907
No	228 (98.7)	71 (98.6)	55 (98.2)	102 (99.0)	
Yes	3 (1.3)	1 (1.4)	1 (1.8)	1 (1.0)	
**Number of vaccine doses, median (IQR)**	3.0 (3.0–3.0)	3.0 (3.0–4.0)	3.0 (3.0–3.0)	3.0 (3.0–3.0)	0.086
**COVID19 severity (WHO criteria), n (%)**					
Asymptomatic	10 (7.6)	0 (.)	5 (11.6)	5 (5.6)	0.203
Mild	88 (66.7)	0 (.)	31 (72.1)	57 (64.0)	
Moderate	24 (18.2)	0 (.)	6 (14.0)	18 (20.2)	
Severe	10 (7.6)	0 (.)	1 (2.3)	9 (10.1)	

Abbreviations: BTIs, breakthrough infections; CAR-T: chimeric antigen receptor T cells; COPD, chronic obstructive pulmonary disease.

**Table 2 vaccines-12-00784-t002:** Median values of anti-RBD IgG and IFN-γ, GMTs of BA.5 nAbs according to study groups across all the time points (only for available samples).

	Total (n = 231)	NoC (n = 72)	BTIs (n = 56)	H (n = 103)	*p*-Value	* BTIs vs. NoC *p*-Value	* H vs. NoC *p*-Value	* H vs. BTIs *p*-Value
**Anti-RBD IgG T0**	212.7 (5.9–979.0)	46.5 (2.5–815.8)	26.1 (1.4–360.5)	474.1 (105.1–1607.5)	**<0.001**		**<0.001**	**<0.001**
**Anti-RBD IgG T1**	1589.1 (1147.2–2485.3)	1497.6 (1112.1–2053.1)	1265.5 (959.4–2677.6)	1833.2 (1245.2–2652.7)	0.174			
**Anti-RBD IgG T2**	666.0 (478.6–1166.7)	640.5 (397.9–790.4)	622.5 (415.1–1322.2)	802.0 (539.9–1283.2)	0.083			
**Anti-RBD IgG T3**	277.8 (182.2–459.1)	224.3 (163.2–337.1)	295.5 (197.7–862.3)	295.1 (190.9–393.2)	0.378			
**Anti-RBD IgG T4**	80.9 (60.9–432.5)	118.6 (26.1–961.9)	141.1 (64.5–432.5)	63.7 (33.2–69.4)	0.366			
**BA.5 nAbs T0**	8.7 (7.2–10.5)	6.3 (5.1–7.7)	5.6 (4.7–6.7)	14.8 (9.9–22.03)	**<0.001**		**<0.001**	**<0.001**
**BA.5 nAbs T1**	19.2 (15.9–23.1)	14.1 (11.0–18.1)	22.2 (14.3–34.5)	21.2 (16.2–27.8)	0.183			
**BA.5 nAbs T2**	9.6 (7.7–12.0)	5.6 (5.1–6.3)	13.4 (7.7–23.4)	11.7 (8.2–16.7)	**0.002**	**0.006**	**0.002**	
**BA.5 nAbs T3**	8.7 (6.3–12.1)	5.2 (4.8–5.6)	17.4 (8.9–33.7)	5.0 (5.0–5.0)	**<0.001**	**<0.001**		**0.002**
**IFN-γ T0**	38.2 (7.0–204.4)	27.5 (4.0–140.0)	26.8 (1.5–180.4)	92.0 (16.1–343.4)	**0.039**		**0.014**	
**IFN-γ T1**	95.0 (14.5–242.5)	52.3 (15.0–183.0)	112.8 (4.0–351.8)	106.3 (22.4–186.4)	0.554			
**IFN-γ T2**	66.2 (5.2–191.2)	47.2 (3.6–131.3)	79.5 (0.3–237.0)	119.2 (14.1–282.5)	0.200			
**IFN-γ T3**	121.8 (41.1–367.8)	25.7 (2.8–133.7)	103.1 (29.1–697.5)	176.3 (100.3–398.9)	0.212			
**IFN-γ T4**	589.0 (216.5–728.5)	216.5 (216.5–216.5)	589.0 (589.0–589.0)	728.5 (728.5–728.5)	0.367			

Data are expressed as the median and interquartile range (IQR); * *p*-value < α/3 for Bonferroni multiple testing correction.

## Data Availability

Data will be available upon reasonable request to the corresponding author.

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
