# Peer review of "Real World Use of Tixagevimab/Cilgavimab Pre-Exposure Prophylaxis of COVID-19 in Immunocompromised Individuals: Data from the OCTOPUS Study"

_vaccines, 2024, doi:10.3390/vaccines12070784_

Round 1

Reviewer 1 Report

Comments and Suggestions for Authors

Vergori and colleagues report on real-world use over time in immunocompromised subjects receiving tixagevimab/cilgavimab pre-exposure prophylaxis for COVID-19 infections.

Major comments:

-          The authors tend to lump all of their immunocompromised patients together, but there are subsets at much higher risk than others. For example, patients in the first year after stem cell transplant for hematologic malignancy are at much higher risk than those persons who are 2 or more years away from their transplant (PMID: 37128256, PMID: 36906276). Similarly, recent Rituxan-treated patients are at high risk. Is the clinical information about this database strong enough to ferret out some of these particularly high-risk patients and add those demographics to Table 1 (Car-T is already listed)? If so, can there be some separate analyses of these patients?

Comments on the Quality of English Language

Minor comments:

-          The abstract is a little hard to follow with some of the abbreviations, which are not explained (RBD, IR, BTIs, GMTs). Please have someone read through it who has not seen it before to assess for flow & some re-writing to make it clearer. I realize you may be working within a word limit that makes it hard.

-          When using the abbreviation TM for trademark, it should be superscripted.

-          There are some spacing issues to address throughout the manuscript. Search for accidental double spaces that should really be single spaces. This includes affiliation 11 on line 29.

-          Many sentences are missing a word or two that would turn them into readable sentences. Please have a native English speaker read each sentence to try to fix these. They are too numerous to elaborate individually. Also, the end of a sentence should have a “.”, such as line 72.

-          You probably need to define COVID-19 (which you also sometimes abbreviate COVID19) and SARS-CoV-2 at least once.

-          When using comparative words, such as “higher”, you need to state higher than _____?

-          Lines 118 to 130, which describe the main study outcomes in a single long sentence, can actually be deleted since each of these items is the title of a separate paragraph in the results section.

-          Line 180: “n. of vaccines doses” should be written out as “number of vaccines doses”.

-          The bibliography has a number of issues that make this reviewer wonder if a reference manager was used. Would double-check the original manuscript that all references are being cited in the correct area.

-          Reference 2 has the page numbers on a new line when that is not needed.

-          DOIs are missing from many of the references

-          References 10 and 11 are web pages that do not report date accessed.

-          PMID, PMCID, and PMC numbers can be removed from references 16, 17, 18, 21, 31, 32, 33, 34, 35, 36, 37

-          Remove “[published online ahead of print …]” from references 20, 22, 24, 29; since these are old references and should have been updated by this point in time.

-          Reference 28 is an abstract from 2023 and should be replaced with the appropriate peer-reviewed publication.

-          Why is reference 30 double-spaced, when the rest are not?

Author Response

Vergori and colleagues report on real-world use over time in immunocompromised subjects receiving tixagevimab/cilgavimab pre-exposure prophylaxis for COVID-19 infections.

Major comments:

-          The authors tend to lump all of their immunocompromised patients together, but there are subsets at much higher risk than others. For example, patients in the first year after stem cell transplant for hematologic malignancy are at much higher risk than those persons who are 2 or more years away from their transplant (PMID: 37128256, PMID: 36906276). Similarly, recent Rituxan-treated patients are at high risk. Is the clinical information about this database strong enough to ferret out some of these particularly high-risk patients and add those demographics to Table 1 (Car-T is already listed)? If so, can there be some separate analyses of these patients?

Thank you for your valuable feedback. We appreciate your suggestion for a stratified analysis for groups of patients at risk. Although none of the participants were in full remission or were off treatment for many months, the exact date of the last immunosuppressive treatment was not available in the database to respond adequately to your request. We will commit to address this important aspect in future research with a sample size that will also consider possible sub-analyses. We will include your suggestions as a limit in the Discussion paragraph.

Minor comments:

-          The abstract is a little hard to follow with some of the abbreviations, which are not explained (RBD, IR, BTIs, GMTs). Please have someone read through it who has not seen it before to assess for flow & some re-writing to make it clearer. I realize you may be working within a word limit that makes it hard.

Thank you for your comment, we have rephrased the abstract trying to make it easier to read and at the same time properly concise. We have also specified where possible all acronyms.

-          When using the abbreviation TM for trademark, it should be superscripted.

Thank you for this specification, we have now amended.

-          There are some spacing issues to address throughout the manuscript. Search for accidental double spaces that should really be single spaces. This includes affiliation 11 on line 29.

Thanks for the comment, we have checked the double spaces and corrected them accordingly.

-          Many sentences are missing a word or two that would turn them into readable sentences. Please have a native English speaker read each sentence to try to fix these. They are too numerous to elaborate individually. Also, the end of a sentence should have a “.”, such as line 72.

Thanks for the comment, we have checked and corrected where absent by mistake fullstops.

-          You probably need to define COVID-19 (which you also sometimes abbreviate COVID19) and SARS-CoV-2 at least once.

Thank you, we specified acronyms in the introduction section.

-          When using comparative words, such as “higher”, you need to state higher than _____?

Thank you for the clarification, .in the results section we specify the comparator to which the adjective higher or lower refers.

-          Lines 118 to 130, which describe the main study outcomes in a single long sentence, can actually be deleted since each of these items is the title of a separate paragraph in the results section.

We have reframed this part of the methods, making the period more readable and concise.

-          Line 180: “n. of vaccines doses” should be written out as “number of vaccines doses”.

Amended, thank you.

-          The bibliography has a number of issues that make this reviewer wonder if a reference manager was used. Would double-check the original manuscript that all references are being cited in the correct area.

Checked and amended, thank you.

-          Reference 2 has the page numbers on a new line when that is not needed.

Amended, thank you.

-          DOIs are missing from many of the references

Thank you, we have now added DOI where appropriate

-          References 10 and 11 are web pages that do not report date accessed.

Amended, thank you.

-          PMID, PMCID, and PMC numbers can be removed from references 16, 17, 18, 21, 31, 32, 33, 34, 35, 36, 37

We have now removed them from refs indicated.

-          Remove “[published online ahead of print …]” from references 20, 22, 24, 29; since these are old references and should have been updated by this point in time.

Updated, thank you.

-          Reference 28 is an abstract from 2023 and should be replaced with the appropriate peer-reviewed publication.

Not yet available publication.

-          Why is reference 30 double-spaced, when the rest are not?

It was a typo, amended, thank you.

Reviewer 2 Report

Comments and Suggestions for Authors

The presented paper is a very reliable and comprehensive analysis of the clinical and laboratory response after the use of T/C monoclonal antibodies. I have no comments on the methodology used, results and conclusions. It may be worth emphasizing that before new monoclonal antibodies or other forms of COVID-19 prevention or treatment appear, better suited to new COVID19 variants, T/C may be an element of post-exposure prophylaxis in patients > 12 years of age and > 40 kg body weight from groups high risk of severe COVID-19, but must be used in double dose?

Author Response

The presented paper is a very reliable and comprehensive analysis of the clinical and laboratory response after the use of T/C monoclonal antibodies. I have no comments on the methodology used, results and conclusions. It may be worth emphasizing that before new monoclonal antibodies or other forms of COVID-19 prevention or treatment appear, better suited to new COVID19 variants, T/C may be an element of post-exposure prophylaxis in patients > 12 years of age and > 40 kg body weight from groups high risk of severe COVID-19, but must be used in double dose?

Thank you for your comment, greatly appreciated.

With regard to your question,we believe that to have greater efficacy it is necessary to at least double the dose, as also shown by the studies cited in the text.

However, to date, we are waiting for new mAbs that can be fully effective against the current variants (see trial results on sipavibart long acting moAb for prevention; available at: https://www.astrazeneca.com/media-centre/press-releases/2024/supernova-trial-met-covid-19-prevention-endpoint.html)

Reviewer 3 Report

Comments and Suggestions for Authors

The manuscript Real world use of Tixagevimab/Cilgavimab pre-exposure prophylaxis of COVID-19 in immunocompromised individuals: data from the OCTOPUS study” by Vergori et al presents data from an observational study in immunocompromised subjects receiving the mAbs tixagevimab and cilgavimab (T/C) as pre-exposure prophylaxis (PrEP) to SARS-CoV-2 and COVID-19 disease, and in addition SARS-CoV-2 vaccination. 

According to the described timeline of events, the patients were in the reviewer`s understanding stratified into three groups: 

·         PrEP, 3 x vaccination and no SARS-CoV-2 infection,

·         PrEP, SARS-CoV-2 infection and 3 x vaccination, and

·         PrEP, 3 x vaccination and SARS-CoV-2 break-through infection.  

Anti-RBD IgG, BA.5 neutralizing antibodies (nAbs), mucosal IgG, and T-cell immunity at the time of T/C administration, and at 3, 6, and 9 months thereafter were measured in these groups.

The authors investigated an important patient cohort in sufficient sample size and the presented data add information on the effects of pre-exposure prophylaxis by mAbs and subsequent vaccination and infection (or vice-versa) on RBD-specific Ab levels, cross neutralization capacity and mucosal Abs. However, the manuscript requires major revisions and clarifications.

Major comments:

Abstract:

Please structure the abstract with paragraphs into sections: background, methods, results and conclusions. Rewrite the abstract to provide a more comprehensive/clearer description of the results in full sentences. Also, the conclusion now states results, which is not the point, please change.

Introduction:

The revised dosing for PrEP by the FDA of 300 mg/300 mg of T/C to ensure the neutralization against the COVID-19 Omicron subvariants is mentioned. The original Wuhan (RBD, S1, S ?) specificity of T/C should be mentioned as well. Also the type of vaccines administered to the study cohort should be provided.

M & M:

Provide a Flowchart and Timeline of events for a better understanding of the investigated groups (as additional figures or in the supplemental material).

The first/general part of the M & M section should be re-structured for easier understanding. Text from line 119 to line 131 is too long for one sentence

Results:

Line 189 to 191 is unclear and should read as follows: …of non-haematological diseases (37/231), 17 were represented by multiple sclerosis, 6 were solid organ transplant, 1 advanced HIV, and 13 were treated with immune-suppressive therapy for autoimmune disorders.

The description of the 3 groups in line 194 to 195 should also include the timing of the SARS-CoV-2 vaccinations that were administered to these patients.

The paragraph 3.2 Anti-RBD IgG and BA.5 neutralizing antibodies persistence is hard to follow and should be rephrased.

Saliva samples are described to be investigated for specific IgA and IgG in the methods section, but only IgG levels are reported. In the discussion, result for IgA are briefly mentioned- these data should also be included in results and possibly a figure.

T cell-specific responses to SARS-CoV-2 spike protein (peptides 1ug/ml; Milteny) were evaluated by measuring IFN-γ production (in re-stimulated PBMC?) by use of an automated ELISA assay – please provide the manufacturer or a description of the in-house assay. Please also provide a reference for the PBMC isolation protocol.  When describing SARS-CoV-2 -specific T-cell responses, please note the correct wording. The expression T-specific responses is unclear.

Indicating the medians of the depicted IFN-γ levels in Figure 3 would improve the information of the graph. Furthermore, showing the results for the 3 groups in one graph would allow to indicate the statistical differences between the groups at T0. Also increase graph size and use the same colour code as in other graphs.

When describing 3.5. Breakthrough infections (BTIs), please refer to Table 1, Table 2 and Figure 1; the described concentrations in the text are not identical with the concentrations given in Table 2.

Discussion:

Please discuss whether measurement of RBD-specific IgG could distinguish between the administered mAbs and the vaccine-induced Abs.

The reviewer also misses a discussion on whether and how much the effectiveness of mAbs to prevent or delay BTI was influenced by presence of vaccine-induced Ab responses. This is needed to put the obtained data in perspective.

Comments on the Quality of English Language

The use of English language needs major editing.

Author Response

The manuscript “Real world use of Tixagevimab/Cilgavimab pre-exposure prophylaxis of COVID-19 in immunocompromised individuals: data from the OCTOPUS study” by Vergori et al presents data from an observational study in immunocompromised subjects receiving the mAbs tixagevimab and cilgavimab (T/C) as pre-exposure prophylaxis (PrEP) to SARS-CoV-2 and COVID-19 disease, and in addition SARS-CoV-2 vaccination.

According to the described timeline of events, the patients were in the reviewer`s understanding stratified into three groups:

  • PrEP, 3 x vaccination and no SARS-CoV-2 infection,
  • PrEP, SARS-CoV-2 infection and 3 x vaccination, and
  • PrEP, 3 x vaccination and SARS-CoV-2 break-through infection.

Anti-RBD IgG, BA.5 neutralizing antibodies (nAbs), mucosal IgG, and T-cell immunity at the time of T/C administration, and at 3, 6, and 9 months thereafter were measured in these groups.

The authors investigated an important patient cohort in sufficient sample size and the presented data add information on the effects of pre-exposure prophylaxis by mAbs and subsequent vaccination and infection (or vice-versa) on RBD-specific Ab levels, cross neutralization capacity and mucosal Abs. However, the manuscript requires major revisions and clarifications.

Major comments:

Abstract:

Please structure the abstract with paragraphs into sections: background, methods, results and conclusions. Rewrite the abstract to provide a more comprehensive/clearer description of the results in full sentences. Also, the conclusion now states results, which is not the point, please change.

Thank you for your comment, we have rephrased the abstract trying to make it easier to read and at the same time properly concise. We have also changed conclusions, as suggested.

Introduction:

The revised dosing for PrEP by the FDA of 300 mg/300 mg of T/C to ensure the neutralization against the COVID-19 Omicron subvariants is mentioned. The original Wuhan (RBD, S1, S ?) specificity of T/C should be mentioned as well. Also the type of vaccines administered to the study cohort should be provided.

We thank the reviewer for this comment, we have now added efficacy data of Evusheld against Wuhan strain in the introduction section.

M & M:

Provide a Flowchart and Timeline of events for a better understanding of the investigated groups (as additional figures or in the supplemental material).

Thank you for this valuable suggestion, we have now added a flowchart with a timeline as supplemental figure 1.

The first/general part of the M & M section should be re-structured for easier understanding. Text from line 119 to line 131 is too long for one sentence

Thank you, it has been rephrased to make it more readable according with the suggestions of all reviewers

Results:

Line 189 to 191 is unclear and should read as follows: …of non-haematological diseases (37/231), 17 were represented by multiple sclerosis, 6 were solid organ transplant, 1 advanced HIV, and 13 were treated with immune-suppressive therapy for autoimmune disorders.

Modified accordingly, thank you for the suggestion.

The description of the 3 groups in line 194 to 195 should also include the timing of the SARS-CoV-2 vaccinations that were administered to these patients.

Unfortunately, we do not have the possibility to determine the exact date of the last vaccination dose.

Evsheld PrEP according with our regulatory agency (AIFA) was possible in those who were found to be unresponsive to the previous vaccine dose or previous COVID19 event.

It was therefore considered at the end of the expected interval for additional/booster doses, which for our regulatory agency was indicated at least 4 months. We have now specified it in the text and added ref.

The paragraph 3.2 Anti-RBD IgG and BA.5 neutralizing antibodies persistence is hard to follow and should be rephrased.

 Thank you, we have now rephrased the entire paragraph.

Saliva samples are described to be investigated for specific IgA and IgG in the methods section, but only IgG levels are reported. In the discussion, result for IgA are briefly mentioned- these data should also be included in results and possibly a figure.

According to referee’s suggestion, we included results in the result section and also added a supplementary figure (Supplemental figure 2)

T cell-specific responses to SARS-CoV-2 spike protein (peptides 1ug/ml; Milteny) were evaluated by measuring IFN-γ production (in re-stimulated PBMC?) by use of an automated ELISA assay – please provide the manufacturer or a description of the in-house assay. Please also provide a reference for the PBMC isolation protocol.  When describing SARS-CoV-2 -specific T-cell responses, please note the correct wording. The expression T-specific responses is unclear.

We thank referee for the clarification. SARS-coV-2 T-cells specific response was evaluated by an Elisa test after stimulating the whole blood of patients with peptides from SARS-CoV-2 Spike protein (peptides 1ug/ml; Milteny). In detail, we stimulated 1 ml of whole blood for each condition (unstimulated, Spike and SEB) for 20 hours at 37°C and 5% of CO2. We tested the IFNg release from SARS-CoV-2 specific T cells in the supernatants of each condition after centrifuging the samples at 2000rpm for 10 minutes.

We tested the IFNg release according to the Elisa kit protocol (Biotechne).

In this study, we didn’t perform this experiment with PBMC from patients.

We changed “T-specific responses” in T-cell specific response according to the referee’s comment.

Indicating the medians of the depicted IFN-γ levels in Figure 3 would improve the information of the graph. Furthermore, showing the results for the 3 groups in one graph would allow to indicate the statistical differences between the groups at T0. Also increase graph size and use the same colour code as in other graphs.

Thank you, we changed the graphs according to your comment.

When describing 3.5. Breakthrough infections (BTIs), please refer to Table 1, Table 2 and Figure 1; the described concentrations in the text are not identical with the concentrations given in Table 2.

Thank you, we have now modified this section by referring to table 2.

Discussion:

Please discuss whether measurement of RBD-specific IgG could distinguish between the administered mAbs and the vaccine-induced Abs.

The assay used cannot distinguish between administered mAbs, vaccine induced, or infection driven IgG. Nevertheless, the participants in this study have non-response or a weak response to previous SARS-CoV-2 vaccination or to COVID-19 as an inclusion criterion at PrEP because of their immunocompromising condition. They may have had a partial response but may have lost that response after time.

Therefore, we may assume that the largest part of the IgG detected from T1 to T3 are from the T/Cs, with the exception of the BTIs population where a quote of the measured antibodies may derive from the infectious event occurred during the follow-up. We have now added it as a study limitation.

The reviewer also misses a discussion on whether and how much the effectiveness of mAbs to prevent or delay BTI was influenced by presence of vaccine-induced Ab responses. This is needed to put the obtained data in perspective

The BTIs IR estimate seems to be consistent since the Poisson model was adjusted also for: number of vaccine doses received and anti-RBD-IgG, thus an interference by these controlling variables is unlikely. We have now added this comment in the discussion section.

Round 2

Reviewer 3 Report

Comments and Suggestions for Authors

Dear authors, please provide a manuscript with tracked changes or write the line numbers, in which the respective chanches were made in the point by point reply - or both.

Otherwise tracking down the changes is too time consuming.

Best regards, Reviewer 1

Comments on the Quality of English Language

I will provide final comments, one the revised version has been resubmitted.

Author Response

Dear authors, please provide a manuscript with tracked changes or write the line numbers, in which the respective chanches were made in the point by point reply - or both.

Otherwise tracking down the changes is too time consuming.

Thank you for pointing out, however we had previously uploaded a version with numbered rows and in track changes, maybe you were sent the clean pdf version (?). In any case, we have now specified the changes in the previous point-by-point reply as follows:

Major comments:

-          The authors tend to lump all of their immunocompromised patients together, but there are subsets at much higher risk than others. For example, patients in the first year after stem cell transplant for hematologic malignancy are at much higher risk than those persons who are 2 or more years away from their transplant (PMID: 37128256, PMID: 36906276). Similarly, recent Rituxan-treated patients are at high risk. Is the clinical information about this database strong enough to ferret out some of these particularly high-risk patients and add those demographics to Table 1 (Car-T is already listed)? If so, can there be some separate analyses of these patients?

Thank you for your valuable feedback. We appreciate your suggestion for a stratified analysis for groups of patients at risk. Although none of the participants were in full remission or were off treatment for many months, the exact date of the last immunosuppressive treatment was not available in the database to respond adequately to your request. We will commit to address this important aspect in future research with a sample size that will also consider possible sub-analyses. We will include your suggestions as a limit in the Discussion paragraph (see lines 475-481)

Minor comments:

-          The abstract is a little hard to follow with some of the abbreviations, which are not explained (RBD, IR, BTIs, GMTs). Please have someone read through it who has not seen it before to assess for flow & some re-writing to make it clearer. I realize you may be working within a word limit that makes it hard.

Thank you for your comment, we have rephrased the abstract trying to make it easier to read and at the same time properly concise. We have also specified where possible all acronyms.

-          When using the abbreviation TM for trademark, it should be superscripted.

Thank you for this specification, we have now amended (see line 97)

-          There are some spacing issues to address throughout the manuscript. Search for accidental double spaces that should really be single spaces. This includes affiliation 11 on line 29.

Thanks for the comment, we have checked the double spaces and corrected them accordingly.

-          Many sentences are missing a word or two that would turn them into readable sentences. Please have a native English speaker read each sentence to try to fix these. They are too numerous to elaborate individually. Also, the end of a sentence should have a “.”, such as line 72.

Thanks for the comment, we have checked and corrected where absent by mistake full stops (see abstract, introduction and methods section)

-          You probably need to define COVID-19 (which you also sometimes abbreviate COVID19) and SARS-CoV-2 at least once.

Thank you, we specified acronyms in the introduction section (see line 96)

-          When using comparative words, such as “higher”, you need to state higher than _____?

Thank you for the clarification, however, we do not quite understand what you are referring to. In the results section we always specify the comparator to which the adjective higher or lower refers (see lines 279-302)

-          Lines 118 to 130, which describe the main study outcomes in a single long sentence, can actually be deleted since each of these items is the title of a separate paragraph in the results section.

We have reframed this part of the methods, making the period more readable and concise (see lines 162-176)

-          Line 180: “n. of vaccines doses” should be written out as “number of vaccines doses”.

Amended, thank you (see line 228)

-          The bibliography has a number of issues that make this reviewer wonder if a reference manager was used. Would double-check the original manuscript that all references are being cited in the correct area.

Checked and amended, thank you.

-          Reference 2 has the page numbers on a new line when that is not needed.

Amended, thank you.

-          DOIs are missing from many of the references

Thank you, we have now added DOI where appropriate

-          References 10 and 11 are web pages that do not report date accessed.

Amended, thank you.

-          PMID, PMCID, and PMC numbers can be removed from references 16, 17, 18, 21, 31, 32, 33, 34, 35, 36, 37

We have now removed them from refs indicated.

-          Remove “[published online ahead of print …]” from references 20, 22, 24, 29; since these are old references and should have been updated by this point in time.

Updated, thank you.

-          Reference 28 is an abstract from 2023 and should be replaced with the appropriate peer-reviewed publication.

Not yet available publication.

-          Why is reference 30 double-spaced, when the rest are not?

It was a typo, amended, thank you.

Round 3

Reviewer 3 Report

Comments and Suggestions for Authors

The manuscript Real world use of Tixagevimab/Cilgavimab pre-exposure prophylaxis of COVID-19 in immunocompromised individuals: data from the OCTOPUS study” by Vergori et al has been thoroughly revised, however, a few issues remain:

·         Line 105 – “other” should be omitted

·         Line 141 - Receptor binding domain

·         Line 209:  This has not been corrected. For a clear understanding the text should be:

“…of non-haematological diseases (16%, 37/231), 17 were represented by multiple sclerosis, 6 were solid organ transplant, 1 advanced HIV, and 13 were treated with immune-suppressive therapy for autoimmune disorders.

·         Figure 3 A now shows IFN-gamma levels at T0. The color code and sequence of groups is not consistent with the other graphs. It should be:

 1st dataset: No Covid - white

2nd dataset: BTI - red

3rd dataset: Hybrid -blue

Also consider using Box and Whisker Plots for better visual analysis.

Comments on the Quality of English Language

Overall, punctuation and spelling still need to be checked.

Author Response

The manuscript Real world use of Tixagevimab/Cilgavimab pre-exposure prophylaxis of COVID-19 in immunocompromised individuals: data from the OCTOPUS study” by Vergori et al has been thoroughly revised, however, a few issues remain:

  • Line 105 – “other” should be omitted

Removed see line 138

  • Line 141 - Receptor binding domain

The typo has been amended, thank you (see line 163)

  • Line 209:  This has not been corrected. For a clear understanding the text should be: 

“…of non-haematological diseases (16%, 37/231), 17 were represented by multiple sclerosis, 6 were solid organ transplant, 1 advanced HIV, and 13 were treated with immune-suppressive therapy for autoimmune disorders.

Thank you, the phrase has been reworded accordingly (see lines 237-240)

  • Figure 3 A now shows IFN-gamma levels at T0. The color code and sequence of groups is not consistent with the other graphs. It should be:

 1st dataset: No Covid - white 

2nd dataset: BTI - red 

3rd dataset: Hybrid -blue

Also consider using Box and Whisker Plots for better visual analysis.

We have modified the Figure 3 accordingly to your suggestion.

Overall, punctuation and spelling still need to be checked

Thank you, checked and amended.